# Sexual Dimorphism in the Chinese Endemic Species *Hynobius maoershanensis* (Urodela: Hynobiidae)

**DOI:** 10.3390/ani12131712

**Published:** 2022-07-01

**Authors:** Huiqun Chen, Rongping Bu, Meihong Ning, Bo Yang, Zhengjun Wu, Huayuan Huang

**Affiliations:** 1Key Laboratory of Ecology of Rare and Endangered Species and Environmental Protection, Guangxi Normal University, Ministry of Education, Guilin 541004, China; chq101526@163.com (H.C.); nmh011088@163.com (M.N.); yb10030821@163.com (B.Y.); 2Guangxi Key Laboratory of Rare and Endangered Animal Ecology, Guangxi Normal University, Guilin 541004, China; 18778395529@163.com

**Keywords:** *Hynobius maoershanensis*, sexual dimorphism, morphological characteristics

## Abstract

**Simple Summary:**

In the present study, we examined the sexual dimorphism of *Hynobius maoershanensis.* The results showed that it exhibits sexual shape dimorphism, with five morphological traits being male-biased and one being female-biased. The observed sexual shape dimorphism between males and females could be explained using the sexual selection and fecundity theory hypotheses.

**Abstract:**

Sexual dimorphism is common in most vertebrate species and has diverse manifestations. The study of sexual dimorphism has critical significance for evolutionary biological and ecological adaptation. In this study, we analysed the morphometric data of *Hynobius maoershanensis*, a rare and endangered species, to examine sexual dimorphism in size and shape. A total of 61 *H*. *maoershanensis* individuals (9 adult females and 52 adult males) were used in this study. We measured 14 morphological variables and weight of each individual. Analysis of covariance using snout–vent length (SVL) as the covariate showed significant differences in head width (HW), tail length (TL), tail height (TH), forelimb length (FLL), hindlimb length (HLL) and space between axilla and groin (AGS) between the male and female. The female AGS was greater than that of the male, whereas males had greater HW, TL, TH, FLL and HLL than females. The findings show that sexual dimorphism is present in terms of shape but not in terms of size. The wider head of the male could improve mating success, and its thicker limbs and longer tail might facilitate courtship. The females’ wider AGS may increase reproductive output. Our results support sexual dimorphism in *H. maoershanensis**,* which could be explained by the sexual selection and fecundity theory hypothesis.

## 1. Introduction

Sexual dimorphism is a pervasive phenomenon in many organisms [1]. This term refers to significant differences in body size [2], colour [3,4] and morphology [5,6] between males and females of the same species. Sexual dimorphism in the body size or weight of sexually mature individuals is called sexual size dimorphism (SSD), whereas that in other morphological characteristics is called sexual shape dimorphism (SShD) [5]. SShD and SSD between males and females are common in amphibians [7,8]. For instance, the phenomenon of limb SShD is common in the Urodela, and in most cases males have longer and stronger limbs than females [9,10]. Reinhard et al. showed that the strong forelimbs of males may be associated with courtship behaviour [11]. In amphibians, 90% of anurans and 61% of urodelans exhibit female-biased SSD [12,13]. Sexual dimorphism in body size is often associated with the reproductive behaviours of animals [14,15]. In amphibians, the larger body size in females may be related to the ability to produce more offspring [16,17], whereas the larger size in males may aid in finding mates or caring for offspring [8,14]. For example, in male-biased SSD poison frogs, males often provide more care than females, such as egg attendance and tadpole transport [14]. Male parental care may predict better modes of male-biased SSD than fighting [18]. The parental care behaviour of amphibians is mainly associated with egg attendance [19,20]. Studies have shown that if rearing offspring can increase the opportunities to mate, it is usually the males who provide the rearing, and they are larger in size than the females [8,21,22]. This male-biased SSD can result from reproductive strategy (i.e., parental care) and intrasexual competition [7].

Sexual dimorphism is the result of multiple selection forces [5,7,13]. Three hypotheses identify and explain the various forces influencing SSD or SShD. Sexual selection is an important driving force for SSD [23,24]. The sexual selection hypothesis posits that sexual selection factors are the main driving force for males to develop in the direction of larger body size [25]. Sexual selection may promote a larger body size in males. Thus, males with a larger body size possess a certain advantage and can win the struggle and obtain the right to mate in mate competition [18]. For example, the pattern in the size of *Sceloporus variabilis* is male-biased; male *S. variabilis* maintains larger dimensions to compete for dominion and attain the right to mate [26]. Similarly, the SSD of *Leptobrachium boringii* is male-biased; by occupying territories, larger males achieve greater mating success using cuticularised maxillary nuptial spines as armament to defend a high-quality territory [8]. Moreover, sexual selection has an effect on female mate selection [27]. For females, selecting larger males is beneficial and may be linked to ensuring offspring quality [27,28].

The fecundity theory hypothesis can also explain SSD or SShD. Fertility selection may be the main driving force for females to fit a larger size [13]. Larger females have higher fitness; thus, larger size is a positively selected trait. A larger body size or abdominal cavity could favour the increase in fertility and reproductive output [16,29]. For instance, *Microhyla fissipes* and *Tylototriton shanjing* are female-biased; females have a larger abdominal cavity to contain an increased number of eggs, which enhances the reproductive output [30,31].

The niche separation hypothesis assumes that the separation and difference in niches between individuals of different sexes is the reason for the appearance of SShD, e.g., in the diet [1]. For example, in *Laticauda colubrina**,* females have larger heads than males of the same body size and feed mainly on larger prey [32]. It is also regarded as a consequence of sexual selection because some structures, such as the head, are related to combat between males and resource utilisation [33,34]. However, there are also species with sexual monomorphisms not supported by any of the three hypotheses, e.g., SSD is not present in the salamander species of both *Salamandra algira* and *Mertensiella caucasica* [11].

*Hynobius maoershanensis* (Amphibia, Urodela, Hynobiidae, *Hynobius*) is known only from the Guangxi Zhuang Autonomous Region Maoershan Nature Reserve in Xing’an County. The species chiefly inhabits alpine swamp areas at an altitude of 1950–2000 m and has a very small wild population (1500–1600 individuals) [35,36]. In February 2021, it was listed as a national Level I key protected wildlife in China [37]. An understanding of the related sexual dimorphism and reproductive behaviour of *H. maoershanensis* is still lacking. Hence, in this study, we aimed to explore the sexual dimorphism in *H. maoershanensis* adults. The study of sexual dimorphism can aid us in comprehending the relationship between the sexual morphological characteristics and reproductive behaviour of this species. Furthermore, the study will deepen our understanding of the morphology and reproductive behaviour of urodelan amphibians and contribute to the development of protective strategies for *H. maoershanensis.* The species has a male egg attendance habit [36]; hence, we predicted that its SSD pattern would be male-biased.

## 2. Materials and Methods

### 2.1. Study Site

We conducted sampling in ditches and pools in the Guangxi Zhuang Autonomous Region Maoershan Nature Reserve in Xing’an County (110°24′–110°26′ E, 25°52′–25°53′ N). The altitude of the main peak is 2141.5 m, and it has a typical subtropical mountain climate. The mean annual temperature at the site is 12.8 °C, with an approximate annual rainfall of 2546 mm, an average air humidity of 92% and a frost period of about 95 days [38].

### 2.2. Data Collection

A total of 61 adult specimens (9 females, 52 males) of *H. maoershanensis* were used in this study. The specimens were collected from the Guangxi Zhuang Autonomous Region Maoershan Nature Reserve in Xing’an County. *H. maoershanensis* individuals were sexed by inspection of the cloacae. The outer wall of the cloaca of the male is obviously protruding, with a slightly ‘M’-shaped transverse anal fissure, whereas the female anal hole is round [36]. We referred to the morphological characteristics according to Fei et al. [39]. The weight (W) of *H. maoershanensis* was measured with an electronic balance (precision of 0.001 g). A digital vernier calliper was used to measure the total length (TOL), head length (HL), head width (HW), snout–vent length (SVL), snout length (SL), trunk length (TRL), interorbital space (IOS), internasal space (INS), tail length (TL), tail height (TH), tail width (TW), forelimb length (FLL), hindlimb length (HLL) and space between axilla and groin (AGS) (precision 0.01 mm) in each individual (Table 1 and Figure 1). The tails of all 61 individuals were intact and used for the measurement. All *H. maoershanensis* specimens were returned to their original location after measurement.

### 2.3. Statistical Analysis

Statistical analysis of the data was performed using SPSS software version 23.0 (SPSS Inc., Chicago, IL, USA). The normality (Kolmogorov–Smirnov) and homogeneity of variance (Levene) were tested before further statistical analysis. When the measured data were homogeneous (i.e., the variances were not significantly different) and follow a normal distribution, linear regression was employed to analyse the relationship between local morphological variables and SVL; if not, the Spearman’s rank test was used. For local morphological variables that exhibited a significant correlation with SVL, a one-way analysis of covariance (ANCOVA) with SVL as a covariate was performed to compare the sex differences. Variables not significantly correlated with SVL were tested for sexual dimorphism using a non-parametric test. Furthermore, an independent samples *t*-test was used to determine the significance of sexual differences in SVL and W. The values were presented as mean ± standard error (SE) of the mean, and statistical significance was assumed at a level of *p* < 0.05. SSD of *H. maoershanensis* was calculated using the size dimorphism index (SDI) according to Gibbons’ [40] formula. The SDI was calculated by dividing the mean SVL of the larger sex by that of the smaller sex and subtracting 1 from it (SVL_larger_/SVL_small__er_ − 1); larger SDI values indicate greater difference between the sexes and vice versa.

## 3. Results

Descriptive statistics showed that the average SVL of females of *H. maoershanensis* was 88.61 ± 2.28 mm (*n* = 9) and that of males was 89.13 ± 0.71 mm (*n* = 52) (see Table 2). The male to female SVL ratio was 1.0058, and the SDI for SVL was 0.0058.

### 3.1. The Relationship between Morphological Eigenvalues and SVL

The mean and SE of *H. maoershanensis* TOL, HL, HW, SVL, SL, TRL, IOS, INS, TL, TH, TW, FLL, HLL, AGS and W are listed in Table 2. The test of homogeneity of variance indicated that the variance of all variables was homogeneous. The normality test results signified that HL, SL, IOS and HLL did not follow a normal distribution. Thus, we used the Spearman’s rank test to assess the correlation between these variables and SVL, and the results implied a significant correlation between HL, HLL and SVL. Additionally, SL and IOS had no significant relationship with SVL (*p* > 0.05). The remaining variables (TOL, HW, TRL, INS, TL, TH, TW, FLL, AGS and W) followed a normal distribution. Therefore, we used the linear regression to analyse the relationship between local morphological variables and SVL. The results alluded a significant correlation between the remaining variables and SVL (see Table 2).

### 3.2. Comparison of the Morphometric Variables of Males and Females

Because SVL demonstrated a high degree of correlation with most variables (*p* < 0.01), an independent samples *t*-test was used to analyse the sexual dimorphism in SVL. W was also analysed using an independent samples *t*-test. The results showed that both SVL (*F*_1,__59_ = 1.162, *p* = 0.285) and W (*F*_1,__59_ = 2.120, *p* = 0.152) did not differ significantly between females and males. The SDI revealed an SSD that was slightly shifted towards males (SDI = 0.0058). Normality test results indicated that HL, SL, IOS and HLL did not follow a normal distribution. Thus, a non-parametric method was employed to test the differences in HL, SL, IOS and HLL between females and males. The findings signified that HL (*p* = 0.404), SL (*p* = 0.188) and IOS (*p* = 0.081) were not significantly different (*p* > 0.05) when the sexual dimorphism was analysed. However, there were significant sex-biased differences in HLL (*p* = 0.001). The remaining variables (TOL, HW, TRL, INS, TL, TH, TW, FLL and AGS) exhibited a significant correlation with SVL. Thus, one-way ANCOVA with SVL as the covariate was used to analyse the sexual dimorphism in these morphological variables. The results implied that the TOL (*F*_1,__59_ = 4.743, *p* = 0.033), HW (*F*_1,60 =_ 8.812, *p* = 0.004), TL (*F*_1,__59_ = 7.250, *p* = 0.009), TH (*F*_1,__59_ = 15.497, *p* = 0) and FLL (*F*_1,__59_ = 22.297, *p* = 0) of male *H. maoershanensis* were significantly greater than that those of the females. Furthermore, females had significantly greater AGS than males (*F*_1,__59_ = 6.796, *p* = 0.012) (see Table 3 and Figure 2).

## 4. Discussion

Sexual dimorphism in the body shape of amphibians can be divided into three types: (a) female adults are larger than male adults, (b) male adults are larger than female adults and (c) there is no difference in the size of male and female adults [5,12]. The first of these is the most prevalent type and favours improvement in reproductive output [12,17,41]. In previous studies, urodelans, such as *Paramesotriton guangxiensis* [42], *Paramesotriton fuzhongensis* [42] and *Cynops orientalis* [43], and anurans, such as *Hyla eximia* [44], *Charadrahyla sakbah* [45], *Crossodactylus schmidti* [46] and *Microhyla beilunensis* [47], among others, have been shown to belong to the first type, i.e., the female adult is larger than the male adult. Amphibians such as *L. boringii* [8] and *Quasipaa spinosa* [48] belong to the second type, i.e., the male adult is larger than the female adult. Both *Batrachuperus taibaiensis* [49] and *Phrynocephalus helioscopus* [50] belong to the third type, i.e., the difference between males and females is not significant. This study showed that, in terms of body size, the difference in the male and female adult *H. maoershanensis* snout–vent length was not significant, and hence, it belongs to the third group.

According to the local feature variables, sexual dimorphism in *H. maoershanensis* was mainly manifested in the body, head, limbs and tail. When the sexes were adjusted for SVL, the male total length, head width, tail length, tail height and limb length were all significantly greater in males than in females, whereas the space between the axilla and groin was significantly greater in females than males. There were significant differences in the total length; however, there was no difference in the snout–vent length. The measurements were performed after excluding the individuals with broken tails; therefore, the difference in the total body length was caused due to the tail length difference. As observed from the analysis, *H. maoershanensis* did not exhibit significant differences in SSD. However, SDI showed a sexual dimorphism that was slightly shifted towards the males [11]. Shine identified sexual selection as the main driving force of males towards larger bodies; the larger size helps males to achieve successful mating [12]. Seglie et al. found that male body length is mainly due to intra-male selection [9]. Sexual selection pressure usually acts on males, because larger males have an advantage in obtaining resources (such as food and space). This advantage affects female selection of males and further provides large males an advantage in sexual selection competition, thereby enhancing their reproductive success [5,11,51].

Most amphibians return to their habitat after oviposition, with no egg attendance [8,9,52]. However, *H. maoershanensis* exhibits the behaviour of long-term egg attendance [36], and its body size is male and female homotype. Some other amphibians, such as *Andrias davidianus* [53], *B. tibetanus* [54], *Allobates subfolionidificans* [55], *L. boringii* [8] and *Andrias*
*j**aponicus* [56], also exhibit egg attendance behaviour. The egg attendance of male *A. davidianus* can greatly improve the hatching rate of fertilised eggs compared with artificial hatching [53]. Therefore, the egg attendance of male *A. davidianus* is crucial for the continuance of the offspring. *B. tibetanus* females exhibit brief egg attendance that can better protect the egg bags and prevent encroachment by other natural enemies or aquatic animals, and are thus most likely to produce more offspring [54]. Of course, these species differ in sexual dimorphism; *B. tibetanus* [54] and *A. subfolionidificans* [55] are female-biased, whereas *L. boringii* [8] is male-biased.

Sexual shape dimorphism in *H. maoershanensis* head size is expressed in terms of the width of male-biased head. Sexual selection, fecundity and ecological selection have been used to explain the dimorphism in head size [5,11,57]. Male-biased head size is generally explained by sexual selection [5]. In competition between males, sexual selection tends to favour individuals with larger heads [58]. Males with larger heads win more easily in male competition and obtain more mating opportunities [5,58]. The wider head is conducive for the intake of a larger quantity of food, which can increase the chance of obtaining high-quality food resources [59], thereby giving them more energy to perform the corresponding activities. The SShD of the *H. maoershanensis* head may be explained using the sexual selection hypothesis.

The tail of *H. maoershanensis* demonstrates SShD (tail length and tail height are male-biased). The tail is an important organ for the urodelans and is related to nutrient storage, natural enemy defence, courtship, respiration and other functions [60,61]. Furthermore, a longer tail is a prominent feature of male courtship [62]. For example, the courting male *Euproctus asper* uses the tail to assist himself in encircling the female’s tail and then to complete the insemination [63]. *H. maoershanensis* exhibits courtship behaviour before amplexus. The courtship process consists mainly of male *H. maoershanensis* holding the branches and then shaking them continuously, swinging its tail to attract the females in the group. The longer tail is conducive to attracting female *H. maoershanensis* [64]. Thus, the longer tail of the male *H. maoershanensis* may serve chiefly to defend against natural enemies, aid in courtship and improve the demands of male reproduction. In addition, energy storage may contribute to the SShD of the tail.

The SShD of *H. maoershanensis* limbs is male-biased. The strong forelimbs of males may be propitious to courtship behaviour [11]. *H. maoershanensis* displays amplexus behaviour [64]. The longer forelimbs of the male can effectively hold females during amplexus so that they cannot run away or be taken away by other males [5,65]. Thus, thick and long limbs improve male courtship ability to ensure smooth amplexus [5]. This sexual dimorphism in the limb is also found in two other urodelan species, namely *Liua shihi* [57] and *Pachyhynobius shangchengensis* [5].

In the case of most morphological variables, males were larger than females; however, there were also female features that were larger than those of males. The space between the axilla and groin relative to the snout–vent length of the female adult *H. maoershanensis* was significantly greater than that of the male. The large space between the axilla and groin could be to accommodate a larger abdominal cavity to produce larger ovaries and large numbers of ova, thereby increasing fertility and reproductive output [31,58]. Similar to most Hynobiidae, the female *H. maoershanensis* have a larger space between the axilla and groin than males, which may confer a fecundity advantage, thereby providing a potential explanation for the fecundity theory.

In summary, the sexual size dimorphism pattern of *H. maoershanensis* was male and female homotype, which demonstrates that our previous speculation on the body size of the *H. maoershanensis* cannot be accepted. The sexual dimorphism in the external morphology was mainly manifested in the head, space between the axilla and groin, tail and limbs. Compared with females, males have a greater head width, tail length, tail height and limb length. The sexual dimorphism of *H. maoershanensis* may be explained by the sexual selection and fecundity theory hypotheses. The currently available data do not support the niche separation hypothesis.

## Figures and Tables

**Figure 1 animals-12-01712-f001:**
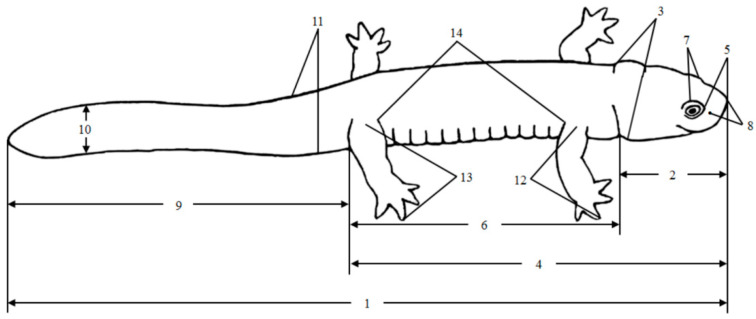
Urodela adult. The morphological variables used in the study (adapted from [39]). Note: 1—TOL, 2—HL, 3—HW, 4—SVL, 5—SL, 6—TRL, 7—IOS, 8—INS, 9—TL, 10—TH, 11—TW, 12—FLL, 13—HLL and 14—AGS.

**Figure 2 animals-12-01712-f002:**
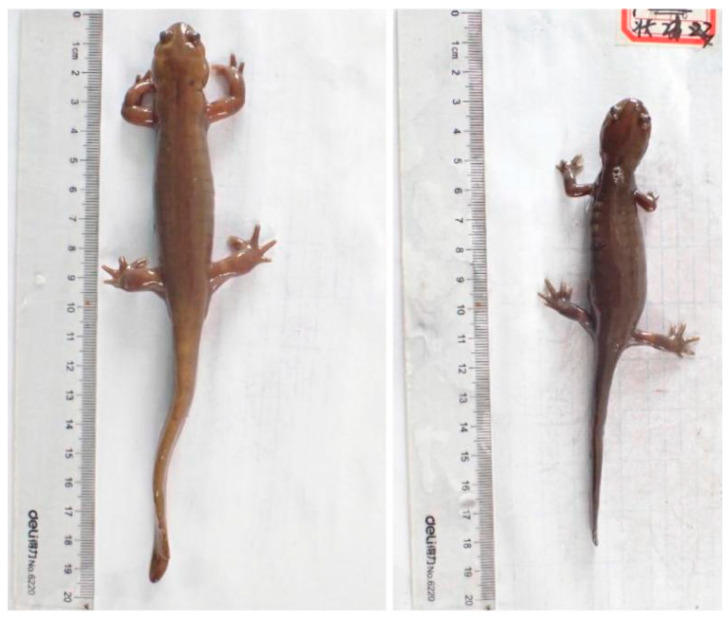
Sexual dimorphism in *Hynobius maoershanensis*. Overall view. Male (**left**) and female (**right**).

**Table 1 animals-12-01712-t001:** Morphological variables assessed in this study and their abbreviations.

Morphological Variables	Definition
1. Total length (TOL)	From the snout to the tail end
2. Head length (HL)	From the rostral end to the posterior edge of the jaw joint
3. Head width (HW)	Width of the head at its widest point
4. Snout–vent length (SVL)	From the tip of the snout to the posterior margin of the cloaca
5. Snout length (SL)	From the anterior border of the eye to the tip of the snout
6. Trunk length (TRL)	Length from the neck drape to the posterior edge of the anal hole
7. Interorbital space (IOS)	The narrowest distance between the inner margins of the left and right upper eyelids
8. Internasal space (INS)	The distance between the two nostrils
9. Tail length (TL)	From the posterior margin of the cloaca to the tip of the tail
10. Tail height (TH)	The height of the tail at its highest point
11. Tail width (TW)	The width of the tail at its widest point
12. Forelimb length (FLL)	Length from the base of the forelimb to the longest finger end
13. Hindlimb length (HLL)	Length from the base of the hindlimb to the end of the longest toe
14. Space between axilla and groin (AGS)	Distance from the posterior edge of the forelimb to the anterior edge of the hindlimb

**Table 2 animals-12-01712-t002:** Descriptive statistics for original morphometric variables (mm/g) in female and male *H. maoershanensis*.

Variables	*Hynobius maoershanensis*		
	Female (*n* = 9)	Male (*n* = 52)	*p*-Value
	Mean ± SE	mean ± SE	
SVL	88.61 ± 2.28	89.13 ± 0.71	-
TOL	160.18 ± 5.63	167.68 ± 1.83	<0.001 **
HL	18.90 ± 1.34	19.82 ± 0.60	0.003 **
HW	15.46 ± 0.69	17.89 ± 0.36	<0.001 **
SL	4.44 ± 0.21	4.77 ± 0.10	0.061
TRL	71.47 ± 2.58	70.52 ± 0.81	<0.001 **
IOS	5.22 ± 0.23	5.74 ± 0.13	0.246
INS	5.24 ± 0.27	5.42 ± 0.07	<0.001 **
TL	70.12 ± 3.69	78.32 ± 1.28	<0.001 **
TH	9.81 ± 0.45	11.74 ± 0.22	<0.001 **
TW	6.61 ± 0.34	7.55 ± 0.26	<0.001 **
FLL	22.24 ± 0.51	25.62 ± 0.29	0.019 *
HLL	25.74 ± 0.36	28.08 ± 0.31	0.007 **
AGS	46.17 ± 1.84	43.02 ± 0.63	<0.001 **
W	20.64 ± 2.36	22.11 ± 0.65	<0.001 **

Note: * = *p* < 0.05; ** = *p* < 0.01.

**Table 3 animals-12-01712-t003:** Analysis of sexual shape dimorphism in *H. maoershanensis*. Results of one-way ANCOVA comparing 10 traits of the sexes relative to SVL.

Variables	*F*-Value	*p*-Value	Sex Bias
TOL	4.743	0.033	M
HW	8.812	0.004	M
TRL	0.557	0.459	n.s
INS	0.693	0.409	n.s
TL	7.250	0.009	M
TH	15.497	0.0001	M
TW	2.533	0.117	n.s
FLL	22.297	0.00001	M
HLL	9.616	0.003	M
AGS	6.796	0.012	F

Note: M, male-biased; F, female-biased; n.s, no sex bias.

## Data Availability

Data are available from the authors upon request.

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
