# Peer review of "Sexual Dimorphism in the Chinese Endemic Species Hynobius maoershanensis (Urodela: Hynobiidae)"

_animals, 2022, doi:10.3390/ani12131712_

Round 1

Reviewer 1 Report

The manuscript entitled “Sexual Dimorphism in the Chinese Endemic Species Hynobius maoershanensis (Urodela: Hynobiidae)” provides novel information on sexual dimorphism of an endemism from China. The study shows that males and females differ in some morphological traits with ecological and reproductive relevance. I think the study is interesting and adds to the knowledge of the species’ ecology and biology, but I think it should be thoroughly revised by the authors before acceptation for publication.

My first comment is about length. I think it is too lengthy and that it could be significantly reduced. Also, it includes too many, and sometimes unnecessary (see following comments), references (95!). Next, I am providing some general comments, and then other more specific.

1.       My main concern is about data analysis, and consequently, the results. I think authors should revise the analysis, or at least, the way they explained them in the material and methods section, which was a bit hard to follow.

For instance, they tested for normality and homocedasticity, but do not specify how they treated deviations from them (e.g. do SVL and W meet the normality/homocedasticity assumption?).

L137-139: did you use linear regression or Spearman test (non-parametric)?

Also, I do not understand what authors meant with sentences such as “Descriptive statistics ± standard error indicate that the significance level is set at α = 0.05.”.

L144-145: this is the same regression as before but differentiating sexes?

To sum up, authors should be careful when explaining methods, but also results, and try to follow a sequential and logical order that facilitates understanding from an external reader.

2.       Results. Another important point that I do not see addressed anywhere is the artifact of differences in Total Body Length (TOL). You did not find SSD in SVL but you did in tail length. Thus, observed differences in TOL are basically due to differences in tail length. Body size in amphibians is usually measured with SVL because the tail is a highly functional structure that are also exposed to many damages (predators, injuries) and can misrepresent the actual size of individuals. I think this should be highlighted anywhere.

3.       In general, I think authors should be more precise when talking about ‘sexual dimorphism’. They repeatedly used it as a synonym of Sexual Size Dimorphism (SSD), which is not the same. The latest is one kind of Sexual Dimorphism, which also includes differences in shape, coloration, etc. So please, specify where it corresponds throughout the text. For example, from the articles you cited, I guess that the second paragraph of the introduction (L51-63) is about SSD but you should precise it.

4.       Abstract: I think the abstract should be revised to improve the contextualization of the study, and clearly state the objectives, main results/patterns observed and conclusions, avoiding some very specific statements such as “The male to female TOL ratio was 1.047, and the sexual 18 dimorphism index was 0.047.” which are not appropriate for an abstract.

5.       Introduction: I think the information provided in this section is appropriate and helps to understand the context of the study, but maybe, it should be summarized and turned around to follow a sensible order and avoid repetitions. For instance, the first and second paragraph provided almost the same information. Also, both examples in lines 80-85 conclude the same, so they could be pulled together. Finally, it would be advisable to deserve at least one separate paragraph to the relationship between reproductive behaviour and sexual dimorphism, as this will determine you main prediction about sexual dimorphism in this species (L105-106).

6.       Table 1: ensure that all measured morphological trait in the first column fits with its definition in the second one

7.       Table 2: maybe it would be better providing p-values, for instance, in brackets to avoid imprecise expressions such as “extremely significant”.

8.       Check numbers of Figures.

9.       In general, I think authors should reduce the number of references. For instance, in the discussion. It is always interesting to include some examples on other biological systems that present similar patterns (or not) as yours. But I think the discussion present an excessive number of references to other species with no apparent reasoning (phylogenetic proximity? Similar ecologies? Distribution?). Maybe you can filter them using any relevant criterion.

10.   Finally, the high number of abbreviations employed complicates the reading of the manuscript. I recommend to try to use some generalizations: e.g. differences in head measures, in limbs, tails, etc. instead of each single measurements, especially in the discussion.

Some minor/specific comments:

-          Check that all specific names are in italics throughout the manuscript.

-          L61: SSD has not been previously defined.

-          L78-79: please, be careful about this kind of “Lamarkist statements”. Females do not voluntarily increase their size because they want to be more fecund, but larger females have higher fitness and thus, larger size is a positively selected trait.

-          L92: add “any of” before “the three hypothesis”.

-          L97: small/reduced wild population

-           L100-101: delete “in male and female”. The use of both sexes is implicit in the study of sexual dimorphism.

-          L109: sampling instead of field experiments

-          L110: what does “small place name” refer to?

-          L1115: 95 d (days?)

-          L122-124: “We referred to the morphological characteristics mentioned in Search and Drawing of Amphibians in China, according to the appearance and various measurement guidelines for urodelan amphibians.” You mean that you took the list of measured traits from this book? Study? If so, it should be cited and included in the reference list.

-          L129: “of individual shape indexes”, did you calculate any index here or just measured each morphological trait?

-          L146-149: it would be advisable to specify the range of values that this index can take, and what a high/small value mean.

-          L151-152: this information is already provided in Table 2.

-          L156-162: different tests should be also indicated in Table 2.

-          L197: no differences in size between sexes cannot be considered a type of sexual dimorphism…it is just no sexual dimorphism.

-          L220: return or leave?

-          L221: there are also non-endangered species with egg-attendance.

-          L263: change “more athletic” by “males present better performance in swimming and walking” or something similar.

-          L269-270: please, be careful about this kind of expressions. It sound as if females increase their size on purpose to increase their reproductive success.

Reviewer 2 Report

The work can be accepted but requires a more complete editing work. On the other hand, some of the statements are speculative and not supported by data or studies. However, it is a necessary contribution to the knowledge of a species listed as endangered. I think the work needs a deep revision of the editing and although I am not an English speaker, I think it also needs a good proofreading.

The work can be accepted after a major review, because there are no data on this species, although there are similar works on closely related species.

Comments:

The scientific names of species in the Abstract must be written in italics

Line 16 We measured 15 morphological characteristic indexes 16 across individuals. You measure 15 morphological variables. Not characteristics.

Line 36. I don’t understand why the references 9 and 10 are related to sentence of line 36.  These references don’t explain reproductive behaviours.

Line 38. References 13 and 14 are not appropriated. Poecilia Reticulata and Amolops ricketti have males smaller than females. Perhaps references 42 and 43 are more appropriate?

Line 41. I think that reference 17 is not needed. Describes i tan novel type of reproductive behaviour. If not, delete it.

Line 47. Reference nº 17 is to much specific compared to the 28, 29. Delete 17.

Lines 51 -60. I Think that they are a repeated information and could be interesting in the discussion but not in the introduction. I think that it is better focusses the information in amphibians. Delete these lines or justify why not.

Line 61. The first time when you use SSD acronym you need explain it. Include: (Sexual Size Dimorphism)

Line 64. Reference 7 is not general to much particular case.

Line 71. The same comment of line 38.

Line 79. A larger body or abdominal cavity is more conducive to fertility and reproductive output.

Rewrite : A larger body size or abdominal cavity could favour the increase of fertility and reproductive output.

Line 80: Why 12? Delete it.

Line 83-85: female individuals have a 83 larger abdominal cavity to increase the number of eggs, resulting in greater reproductive output and ultimately improving the reproductive success rate

Sure? It is a finalist conclusion. Selection favours larger body size but not larger body size is cause it. I suggest to rewrite as:

female individuals having larger abdominal cavity increase the number of eggs, resulting in greater reproductive  output.

and ultimately improving the reproductive success rate delete it. It is not consequence of higher number of eggs. It depends also of ecological, physiological and inter and intraspecific stressors.

Line 100-101: We aimed to explore specific sexual dimorphism in male and female H. Maoershanensis.

Rewrite: We aimed to explore sexual dimorphism of adults of H. Maoershanensis.

Mat & Met:

Line 115: 95 d. 95 days

Table 1.- Definitions of the morphological characteristic sets and abbreviations.

Better: Table 1.- Morphological variables and abbreviations used in the study.

Table 1.-

After Trunk length (TRL) description all variables’ descriptions are not aligned with de variable name. Perhaps it would be much easier to understand with a figure. I propose to replace Table 1 with a figure indicating the variables taken. Weight (W) is not needed because of it is explained in the text.

Line 135: The normality (Kolmogorov-Smirnov) and homogeneity of variance (Levene) were tested before further statistical analysis. But the results nota re indicate in Mat & Met. All adjust to normality? In biometric data usually yes, but...

Line 142. Features with no significant correlation were examined using a non-parametric test for sexual dimorphism.

Features= variables?  What non-parametric test?

Results:

Lines 151-154. You must reference Table 2 in the text

Lines 157-161:  Linear regression showed 157 that TOL, HW, TRL, ND, TL, TH, TW, FLL, AGS, and W all had significant or extremely 158 significant relationships with SVL. The remaining features HL, HLL, SL and IOS were analysed by the Spearman test, and the results indicated an extremely significant correlation between HL, HLL, and SVL. Additionally, SL and IOS had no significant relationship 161 with SVL (P > 0.05)

I don’t like extremely significant. Correlations are significant or not. On the other hand, features and characteristic must be replaces by “VARIABLES” You work with variables that are numerical representation and variation of the characteristics and features.

Linear regression for TOL, HW,.... is not also estimated with Spearman test?

Table 2:

Correlation coefficient with SVL(r):  With males, females or all individuals. I suppose that is the last option. Indicate it, please.

Note: * indicates significant correlation, ** indicates extremely significant correlation. Rewrite: * = P<0.05;  ** = P<0.01 (it’s true?)

Line 173-178: Indicates the Post-Hoc test used.

Line 178-180. It is the same information showed in the previous lines. Delete it. And move reference of  Figure 1 to line 174 or 175.

Line 186… You need a test of slopes comparisons to ensure if one is greater than other.

Figure 1. Unary linear regression analysis of SVL and local features of H. Maoershanensis.  Could be showed with a table and only includes the two regression mor important in Figure 1.

Discussion:

Line 199 Paramesotriton guangxiensis in italics. All scientific names in italics. Correct it.

Line 202: [73] all belong to the first type. Rewrite: [73] between others, belong to the first type;

Line 207: than that of the female and that it Rewrite: in females and it ...

Line 211: were all significantly greater than those of females, while AGS in female adults 211 was significantly greater than that of males.

Rewrite: were all significantly greater than females, while AGS in female adults was significantly greater than males.

Line 221.. However, many endangered or vulnerable amphibians have egg attendance behaviour, including Andrias davidianus [17], B. tibetanus [82], Allobates subfolionidificans  [83], L. boringii [8], and A. japonicus [84].

I don’t understand the relation with parenteral care and status of conservation. Other nonendangered species show also parenteral care. Delete it please.

 Line 224: the quality and hatching rate of fertilised eggs.

Hatching rate OK. But quality? What is eggs Quality? . Explain or delete.

Line 228: Similarly, H. maoershanensis has the habit of long-term egg attendance. Have?

Lines 220-229 and all the discussion: If you present in first time one species, you must write the complete scientific name.

Lines 251-253: Females invest more energy into egg production, resulting in a shorter tail, and males invest less into sperm production, resulting in the longer tail [80].

Is too much speculative this sentence. Delete it. it is not believable. I can’t read reference 80, but say it this?

Line 262: The longer forelimbs of males indicate that males are  more athletic than females [55].

Delete it. Humanized comment.

Line 265: The AGS of the female. Rewrite: The AGS related to SVL of the female or it is also significant without covariate SVL?

Line 268: to produce larger ovaries and large num-267 bers of oocysts, Oocyst is a cyst containing a zygote formed by a parasitic protozoan.

You can use oocytes

lines 268-271. Female H. 268 maoershanensis may be affected by reproductive pressure, causing females to constantly 269 increase their reproductive output capacity by increasing AGS to achieve reproductive 270 benefits.  Too much speculative. It is not referenced.  Delete it. The following paragraph explain it better.

Line 283:  speculation not. Hypothesis. The same in introduction.

Line 284: was correct.  Can be accepted.

“References

They are many scientific names without italics. Correct it.

Line 438. (accession date)

Round 2

Reviewer 1 Report

The revised version of the manuscript entitled “Sexual Dimorphism in the Chinese Endemic Species Hynobius maoershanensis (Urodela: Hynobiidae)” has greatly improved. Authors have done a really good job integrating many of the reviewer’s suggestions, and now the contextualization and especially, analysis, are much clearer and easy to follow. However, I still have some comments and doubts that I hope may improve the quality of the manuscript before being accepted for publication in Animals:

Abstract

The abstract has been significantly improved, and now it is much more informative and provides a good genera idea of the study and main results. Some minor comments:

L11: in the Simple summary you stated that all variables are male-biased, but AGS is female-biased.

L15: delete “the” before “morphometric data”

L18-20: “There was a significant 18 difference in total length (TOL) between sexes. Analysis of covariance using snout–vent length 19 (SVL) as the covariate showed significant differences in TOL,”

Maybe this is quite redundant

Introduction

L33-34: body colour [3,4] and body local morphol-33 ogy [5,6]

replace by

colour, and morphology

L35: “individuals” instead of “bionts”?

L44-56: I think this paragraph should be summarized, and even merged to the previous one as they provide the same information. Also, it provides some non-relevant information (for this study, I mean) about parental care, but only the first half of the paragraph relates it with SSD, which is the focus of the study. Finally, you mention here and in following paragraphs that male-biased SSD can be also related to intrasexual competition for mate access. I think you should state it clearer: male-biased SSD can result from reproductive strategy (i.e. parental care) and intrasexual competition.

L47: Add SSD between “male-biased” and “poison frogs”

L61-62: “can promote the development of males in the di-61 rection of larger body size” better “may promote larger body sized in males”?

Material and Methods

L107: Change subsection name: Data collection

L109: I appreciate you include this sentence “The tails of all the 61 individuals were intact and used for measurement” but I think it should be placed after the description of morphological variables (L121).

L113-115: Sorry, but I think this reference should be cited after mentioning the variables (L121), and that there is no need to use such a long sentence, just the reference is enough. Maybe you can use some statements such as “according to” or “adapted from” [47].

Figure 1: Very useful figure! I think it can be improved including in the caption a reference to table 1, or include a list of number:acronym for each variable (I think the latter is the best option).

L127: Change subsection name: Statistical analysis.

L135-136: “Variables with no significant correlation were examined using a non-parametric test for sexual dimorphism.” You mean variables no significantly correlated with SVL, don’t you? But I do not understand the reason why you are using non-parametric test with those variables. Do all of them deviate from normality and homocedasticity? In any case, you should specify which analysis you used to compare those traits between sexes.

Results

L144: section title: Results

Table 2: I suggest to place the variable SVL in the first row, and no including the correlation coefficient as it is meaningless.

L163: to test…what? Specify the test used.

L165: sex-BIASED

Table 3: Check TH and FLL P-values.

Discussion

In general I would call for caution in no repeating information that has been previously given in the introduction section.

L195: tail (without s)

L210-220: In order to highlight your results,  I think you can state the relationship between SSD and parental care in the studied species earlier in the paragraph, and then, compare with the patterns observed in other systems.

L253-254: ova

L262: delete etc.

Reviewer 2 Report

Dears,

After reading the new version of the article, the authors have included most of my suggestions or have justified why they have not done so. For this reason, I consider that the article can be accepted in its current form.

Sincerely

Albert Montori

Author Response

Thank you again for your positive comments and valuable suggestions for the previous revision of the manuscript.